# Physical activity is inversely associated with overall cancer risk among college students in the United States: Results from the National College Health Assessment

**Shenghui Wu** [ID]\*, **Martie Thompson, Adam Hege, Richard W. Christiana** [ID], **Jennifer Schroeder Tyson** [ID]

Department of Public Health and Exercise Science, Beaver College of Health Sciences, Appalachian State University, Boone, NC, United States of America

\* wus@appstate.edu

## Abstract

To our knowledge, this is the first epidemiologic study to examine the association between physical activity (PA) and cancer using data from the American College Health Association-National College Health Assessment (ACHA-NCHA). The goal of the study was to understand the dose-response relation between PA and cancer, as well as the associations between meeting US PA guidelines and overall cancer risk in US college students. The ACHA-NCHA provided self-reported information on demographic characteristics, PA, body mass index, smoking status, and overall cancer during 2019–2022 (n = 293,682; 0.08% cancer cases). To illustrate the dose-response relationship, a restricted cubic spline logistic regression analysis was used to evaluate the association of overall cancer with moderate-to-vigorous PA (MVPA) on a continuous basis. Logistic regression models were used to calculate odds ratios (ORs) and 95% confidence intervals for the associations between meeting the three U.S. PA guidelines and overall cancer risk. The cubic spline observed that MVPA was inversely associated with the odds of overall cancer risk after adjusting for covariates; a one hour/week increase in moderate and vigorous PA was associated with a 1% and 5% reduced overall cancer risk, respectively. Multivariable-adjusted logistic regression analyses showed that meeting the US guidelines for aerobic PA for adults ($\geq$150 minutes/week of moderate aerobic PA or $\geq$75 minutes of vigorous PA) (OR: 0.85), for PA for adults ($\geq$2 days of muscle strengthening activity in addition to aerobic MVPA) (OR: 0.90), and for highly active adults ($\geq$2 days of muscle strengthening activity and $\geq$300 minutes/week of aerobic moderate PA or 150 minutes/week of vigorous PA) (OR: 0.89) were statistically significant and inversely associated with cancer risk. MVPA, especially meeting US guidelines, may be inversely associated with overall cancer among college students in the US. To reduce cancer risks, multilevel interventions to promote US physical activity guidelines among college students are warranted.

**Data Availability Statement:** The data that support the findings of this study are available from the American College Health Association, but

restrictions apply to the availability of these data which are not publicly available. Please see the contact information of the American College Health Association below: American College Health Association National College Health Assessment Program Office 8455 Colesville Road, Suite 740 Silver Spring, MD 20910 P: (410) 859-1500 F: (410) 859-1510 www.acha.org.

**Funding:** The authors received no specific funding for this work.

**Competing interests:** NO authors have competing interests.

## Introduction

In the United States (U.S.), approximately 41 out of 100 men and 39 out of 100 women will develop cancer during their lifetime [1]. In 2023, there is estimated to be more than 1.9 million new cancer cases diagnosed and 609,820 cancer deaths in the U.S [1]. The role of physical activity (PA) in cancer incidence are marked, and a large number of cancer cases and deaths could be prevented with the adoption of healthier lifestyles, including not smoking, maintaining healthy body weight, and being physically active [2]. Although epidemiologic studies have shown that inadequate PA was associated with many sites of cancer [3–6] (such as an inverse association of leisure time PA [4] and household PA [6] with cancer risk, and a negative association between PA and cancer mortality in both the general population and cancer survivors [5], a few gaps in research remain and further research is needed to determine the diverse amounts and intensities of PA required for cancer prevention and survival [3]. The U.S. Department of Health and Human Services and the American Cancer Society recommend that adults should get 150–300 minutes of moderate-intensity or 75–150 minutes of vigorous intensity aerobic activity each week (or a combination) [2,4–7].

The college student population in the U.S. is growing [8] and forms an gradually important portion of the U.S. community. Total fall enrollment in degree-granting postsecondary institutions in 2021 was 18,961,280, and it is projected to be 20,233,776 in 2031 [8]. Therefore, the research focused on understanding or changing health risk behaviors of college students, such as lack of PA is very important. However, less than 50% of college students meet PA guidelines [9], and the prevalence of chronic diseases (such as cancer, diabetes, auto-immune disorder) is 6.4% in college students [10]. To our knowledge, there are no epidemiological studies to examine the association between PA and cancer risk among U.S. college students. To understand the dose-response relation between PA and cancer, as well as the associations between meeting U.S. PA guidelines and overall cancer risk in U.S. college students, we examined these associations using data from the American College Health Association-National College Health Assessment (ACHA-NCHA) in the U.S.

## Materials and methods

### Data source and sample

The ACHA-NCHA is a nationwide survey conducted with college students that collects data on a wide range of health and health-related behaviors. Data were obtained from the 2019 to 2022 administration of the ACHA-NCHAIII [11], which captured cross-sectional web-based self-reported information from college students in the U.S. Institutions. Schools that self-selected to participate conducted the survey to a random sample of enrolled students aged 18 years or older. The national database consists of only the data from institutions that randomly selected students or classrooms for participation. Considering the schools are self-selecting, the data limit the generalizability to the U.S. population of college students and schools; nonetheless, a previous evaluation of generalizability found the data to be reliable and valid for representing U.S. students overall, in comparison with other representative samples of U.S. students [12]. This study was exempt from the Appalachian State University Institutional Review Board review as the data were de-identified.

### Exposure and outcome measurements

Collected data included demographic information (such as age, sex, race, ethnicity, health insurance, education, etc.), weight, height, tobacco or nicotine products ever used, moderate and vigorous PA (total minutes) and muscle strengthening activities in the last seven days.

Moderate aerobic PA $\geq$ 150 minutes/week or vigorous PA $\geq$ 75 minutes/week was considered meeting the U.S. aerobic PA guidelines; muscle strengthening activity $\geq$ 2 days in addition to aerobic PA was considered meeting the US PA guidelines for adults; and muscle strengthening activity $\geq$ 2 days and moderate aerobic activity $\geq$ 300 minutes/week (or vigorous activity $\geq$ 150 minutes/week) was considered meeting the U.S. PA guidelines for highly active adults [7]. One minute (hour)/week of vigorous PA is equivalent to two minutes (hours)/week of moderate PA. Cancer diagnosis was identified in participants as being told by a healthcare provider that they had cancer. This study only collected data for the overall cancer but not specific types of cancer. Body mass index (BMI) was calculated by weight in kilograms divided by height squared in meters (kg/m$^2$).

## Statistical analysis

Chi-square test was used to compare participant characteristics for categorical data. To illustrate the dose-response relationship between PA and overall cancer risk, we used a restricted cubic spline logistic regression analysis to evaluate the odds ratios (ORs) of the association of overall cancer with moderate or vigorous PA (hours/week) on a continuous basis [13]. The extreme values (95[th] percentile of the distribution of PA) were excluded to minimize the influence of outliers. Knots were placed at the 5[th], 50[th], and 95[th] percentiles of the distribution of PA. Logistic regression models were used to calculate univariate and fully adjusted ORs and 95% confidence intervals (CIs) for the associations between meeting the three U.S. PA guidelines and overall cancer risk. All models adjusted for age, sex, race, ethnicity, education, BMI, and smoking status. The differences between ORs for the three PA guidelines were examined by using the method described by Allison [14]. A $P$-value $< 0.05$ was considered statistically significant. All statistical analyses were performed using SAS version 9.4 (Cary, NC, USA).

## Results

Among 293,682 participants, 80.04% were 25 years or younger; 67.70% were female. A total of 68.03% of the participants met the U.S. aerobic PA guidelines, 41.96% met the U.S. PA guidelines for adults, and 33.06% met the US PA guidelines for highly active adults. Among all participants, 2,475 (0.08%) reported cancer, and 291,207 (99.82%) did not report cancer (**Table 1**). Compared with those without cancer, participants with cancer were more likely to be older, female, whites, non-Hispanic, undergraduates, cigarette smoking, overweight/obese, and less likely to meet the recommended U.S. PA guidelines (all $P$s $< 0.05$).

**Fig 1** visually depicts the dose-response relationship between moderate (Fig 1A) or vigorous (Fig 1B) PA and the overall cancer risk after adjusting for age, sex, race, ethnicity, education, BMI, and smoking status in a restricted cubic spline model. Total hours of moderate/vigorous PA per week was inversely associated with the risk for overall cancer ($P$ for overall relation = 0.01 and $P$ for linear relation = 0.02). The statistically significant linear inverse dose-response association indicated that for each one hour/week increase in moderate PA and vigorous PA was associated with a 3% [0.97 (95% CI: 0.96–0.99)] and 5% [0.95 (95% CI: 0.91–0.99)] reduced overall cancer risk, respectively.

The ORs were estimated by using the restricted cubic-spline logistic regression models with knots placed at the 5[th], 50[th], and 95[th] percentiles of moderate or vigorous PA. The model was adjusted for age, sex, race, ethnicity, education, body mass index, and smoking status.

Meeting PA guidelines of 150 moderate or 75 vigorous minutes per week was inversely associated with a 22% (OR: 0.78; 95% CI: 0.72–0.85) reduced cancer risk without adjusting for covariates and a 15% reduced cancer risk (OR: 0.85; 95% CI: 0.78–0.93) after adjusting for age, sex, race, ethnicity, education, BMI, and smoking status compared with not meeting guidelines

**Table 1. Characteristics stratified by cancer status: American College Health Association National College Health Assessment (2019–2022).**

| Characteristics | Cancer status (number, percentage) | | | P value |
|---|---|---|---|---|
| | Yes (n = 2,475; 0.84%) | No (n = 291,207; 99.16%) | Total (n = 293,682) | |
| Age (years)[b] | | | | |
| ≤25 | 1,186 (46.97) | 235,519 (80.32) | 236,705 | <0.0001 |
| >25 | 1,339 (53.03) | 57,700 (19.68) | 59,039 | |
| Gender | | | | 0.02 |
| Female | 1,633 (69.91) | 188,744 (67.68) | 190,377 | |
| Male | 703 (30.09) | 90,115 (32.32) | 90,818 | |
| Race | | | | <0.0001 |
| White | 1,717 (75.11) | 177,351 (67.74) | 179,068 | |
| Asian | 215 (9.41) | 47,424 (18.11) | 47,639 | |
| Black | 122 (5.34) | 17,442 (6.66) | 17,564 | |
| American Indian or Native Alaskan | 105 (4.59) | 6,078 (2.32) | 6,183 | |
| Others | 127 (5.56) | 13,522 (5.16) | | |
| Ethnicity | | | | |
| Hispanic | 325 (12.87) | 44,189 (15.07) | 44,514 | 0.002 |
| Non-Hispanic | 2,200 (87.13) | 249,030 (84.93) | 251,230 | |
| Education | | | | |
| Undergraduate | 1,465 (58.32) | 222,832 (76.09) | 224,297 | <0.0001 |
| Master's and above degrees | 917 (36.50) | 66,111 (22.58) | 67,028 | |
| Other | 130 (5.18) | 3,892 (1.33) | 4,022 | |
| Insurance | | | | 0.07 |
| Yes | 2,406 (97.02) | 277,496 (96.82) | 279,902 | |
| No | 74 (2.98) | 9,121 (3.18) | 9,195 | |
| Ever smokers | | | | |
| Yes | 993 (39.45) | 97,238 (33.22) | 98,231 | <0.0001 |
| No | 1,524 (60.55) | 195,437 (66.78) | 196,961 | |
| Body mass index (kg/m²) | | | | |
| Normal (<25) | 1,274 (50.46) | 182,490 (62.24) | 183,764 | <0.0001 |
| Overweight/obese (≥25) | 1,251 (49.54) | 110,729 (37.76) | 111,980 | |
| Met the US guidelines for only aerobic PA for adults[a] | | | | |
| Yes | 1,564 (62.84) | 197,058 (68.08) | 198,622 | <0.0001 |
| No | 925 (37.16) | 92,410 (31.92) | 93,335 | |
| Met the US guidelines for PA for adults[b] | | | | |
| Yes | 935 (37.57) | 121,541 (41.99) | 122,476 | <0.0001 |
| No | 1,554 (62.43) | 167,881 (58.01) | 169,435 | |
| Met the US guidelines for PA for highly active adults[c] | | | | |
| Yes | 708 (28.45) | 95,798 (33.10) | 96,506 | <0.0001 |
| No | 1,781 (71.55) | 193,624 (28.45) | 195,405 | |

PA: Physical activity.

[a] ≥150 minutes/week of moderate aerobic PA or ≥75 minutes of vigorous PA.

[b] ≥2 days of muscle strengthening activity in addition to aerobic PA.

[c] ≥2 days of muscle strengthening activity and ≥300 minutes/week of moderate aerobic PA (or ≥150 minutes/week of vigorous aerobic PA).

(**Table 2**). Multivariable-adjusted logistic regression models showed that meeting PA for adults (≥ 2 days of muscle strengthening activity in addition to aerobic PA), and for highly active adults (≥ 2 days of muscle strengthening activity and ≥300 minutes/week of moderate aerobic

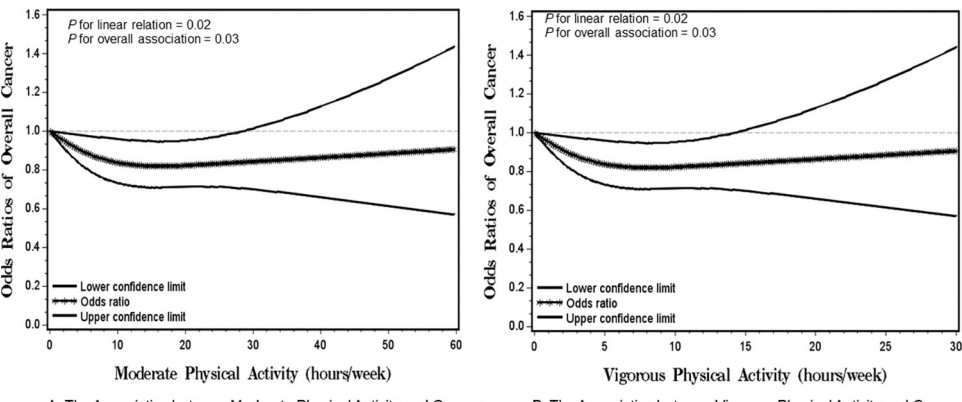

**Fig 1. Smoothed plot for Odds Ratios (ORs) of the overall cancer risk according to moderate or vigorous physical activity (PA) (hours/week).** A. The Association between Moderate Physical Activity and Cancer. B. The Association between Vigorous Physical Activity and Cancer.

PA) was inversely associated with a 10% (OR: 0.90; 95% CI: 0.82–0.98) and a 11% (OR: 0.89; 95% CI: 0.81–0.98) reduced overall cancer risk, respectively. The differences between ORs for the three PA guidelines were not statistically significant (all $P$s>0.05).

## Discussions

Among 293,682 U.S. college students, moderate or vigorous PA was inversely associated with a reduced risk of overall cancer; one hour/week increase in moderate PA and vigorous PA was associated with a 1% and 5% reduced overall cancer risk, respectively. Meeting the US guidelines for aerobic PA for adults (≥150 minutes/week of moderate aerobic PA or ≥75 minutes of vigorous PA), for PA for adults (≥2 days of muscle strengthening activity in addition to aerobic PA), and for highly active adults (≥2 days of muscle strengthening activity and ≥300 minutes/week of moderate aerobic PA) were inversely associated with a 15%, 10%, and 11% reduced overall cancer risk, respectively, after adjusting for age, sex, race, ethnicity, education, BMI, and smoking status. These data strongly indicate the potential for PA to have a deep effect on reported overall cancer risk, and they are more notable because there seems to be a dose-response effect with moderate or vigorous PA as well as exceeding the PA guidelines being associated with lower cancer risk.

**Table 2. Associations between physical activity and cancer: American College Health Association National College Health Assessment (2019–2022).**

|  | Univariate Analysis | | Multivariable adjusted Analysis[a] | |
|---|---|---|---|---|
|  | OR (95%CI) | *P* | OR (95%CI) | *P* |
| Met the US guidelines for only aerobic PA for adults[b] | 0.78 (0.72–0.85) | <0.0001 | 0.85 (0.78–0.93) | 0.0005 |
| Met the US guidelines for PA for adults[c] | 0.83 (0.77–0.90) | <0.0001 | 0.90 (0.82–0.98) | 0.02 |
| Met the US guidelines for PA for highly active adults[d] | 0.80 (0.74–0.88) | <0.0001 | 0.89 (0.81–0.98) | 0.01 |

OR: Odds Ratios; CI: Confidence Intervals; PA: Physical activity.

[a]Adjusted for age, sex, race, ethnicity, education, body mass index, and smoking status.

[b] ≥150 minutes/week of moderate aerobic PA or ≥75 minutes of vigorous PA.

[c] ≥2 days of muscle strengthening activity in addition to aerobic PA.

[d] ≥2 days of muscle strengthening activity and ≥300 minutes/week of moderate aerobic PA (or ≥150 minutes/week of vigorous aerobic PA).

To our knowledge, although no study examined the association between PA and cancer in U.S. college students, one study with a total of 193 French college students indicated that PA could be an effective way to diminish cancer stereotypes and decrease prejudicing against cancer patients (all $Ps < 0.05$) [15]. We found inverse associations between PA and overall cancer risk, especially meeting the three U.S. PA guidelines, were associated with reduced risk of overall cancer. Although differences of inverse associations between meeting the three U.S. PA guidelines and overall cancer risk were not statistically significant, the cubic spline showed that each one hour per week of vigorous PA was associated with a 5% reduced risk of overall cancer while each one hour per week of moderate PA was associated with a 1% reduction in overall cancer risk. It indicates that vigorous PA might bring more beneficial effects than moderate PA regarding the reduction in overall cancer risk. We further found that the multivariable-adjusted inverse association between muscle strengthening PA ($\geq 2$ vs. $< 2$ days/week) was not statistically significant with overall cancer risk (OR: 0.93; 95% CI: 0.85–1.02; $P = 0.12$). It indicated that aerobic moderate or vigorous PA part of the guidelines provides the most benefit in terms of risk reduction compared to the muscle strengthening part. When sample size for meeting three PA guidelines became larger, we may have an opportunity to further examine associations between meeting three PA guidelines and cancer risk among college students. The underlying mechanisms are still mostly unknown for different sites of cancer; however, some potentially PA-modulated parameters, such as insulin sensitivity (e.g., insulin levels, C-peptide, and insulin like growth factor 1), the immune system (e.g., natural killer cell cytotoxic activity, total lytic units, and spontaneous lymphocyte proliferation), and inflammation [e.g., C-reactive protein (CRP) and serum amyloid A] appear to be involved in tumor development [16–18].

There are several limitations in our research. In this cross-sectional study, we only examined an association but not a cause-and-effect relationship because both were accessed at the same time. We are unable to investigate whether cancer came after PA in time or exercise level was caused by cancer; thus, we are unable to provide evidence of a temporal relationship between PA and cancer. Prospective cohort studies and clinical trials are needed to further explore the mechanisms and the dose-response relationship between PA and cancer risk. We used self-reported data to identify cancer status, and hence we are unable to confirm the cancer diagnosis; however, one study has validated the self-reported method [19], and another study suggested that self-reported cancer diagnoses in the U.S. Health Retirement Study showed rational validity which can be used in population-based research that is maximized with linkage to Medicare [20]. We could not examine the associations between PA and risk for cancer in specific sites because the information on cancer specific sites was not collected. The presence of residual confounding due to unmeasured or insufficiently/misclassified collected covariates cannot be completely excluded, but a few potential confounders were adjusted for in our study.

This study has some strengths. First, to our knowledge, this is the first epidemiological study to examine the association between PA and cancer among the U.S. college students. Specifically, we identified the dose-response relationships between PA and overall cancer risk, which provided the data for future research related to cancer prevention. Second, this is a large random sample, hence reducing bias intrinsic in studies taking samples from non-random populations. Third, we can conduct a relatively thorough analysis including pertinent factors because the information on PA and other multiple factors related to cancer was available.

In conclusion, increased moderate and vigorous PA, especially meeting U.S. recommended guidelines, was significantly associated with the reduced risk of overall cancer after excluding the effect of other confounding factors despite limitations of the cross-sectional study design and possible residual confounding. Therefore, PA might be a modifiable protective factor for

which college students can make changes to reduce their cancer risk. Efforts need to be focused on improving multilevel interventions to promote MVPA among college students. Future prospective cohort studies and clinical trials are warranted to further confirm our findings.

## Acknowledgments

The opinions, finds, and conclusions reported in this article are those of the authors, and are in no way meant to represent the corporate opinions, views, or policies of the American College Health Association (ACHA). ACHA does not warrant nor assume any liability or responsibility for the accuracy, completeness, or usefulness of any information presented in this article.

## Author Contributions

**Conceptualization:** Shenghui Wu.

**Data curation:** Martie Thompson.

**Formal analysis:** Shenghui Wu.

**Methodology:** Shenghui Wu.

**Project administration:** Shenghui Wu.

**Resources:** Martie Thompson.

**Software:** Shenghui Wu.

**Supervision:** Shenghui Wu.

**Writing – original draft:** Shenghui Wu.

**Writing – review & editing:** Shenghui Wu, Martie Thompson, Adam Hege, Richard W. Christiana, Jennifer Schroeder Tyson.

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
