## [Decision Letter · Decision Letter 0]

11 Apr 2023

PONE-D-23-06157Physical Activity is Inversely Associated with Overall Cancer Risk Among College Students in the United States: Results from the National College Health AssessmentPLOS ONE

Dear Dr. Wu,

Thank you for submitting your manuscript to PLOS ONE. After careful consideration, we feel that it has merit but does not fully meet PLOS ONE’s publication criteria as it currently stands. Therefore, we invite you to submit a revised version of the manuscript that addresses the points raised during the review process.

We look forward to receiving your revised manuscript.

Kind regards,

Bojan Masanovic, Ph.D.

Academic Editor

PLOS ONE

Journal Requirements:

2. Thank you for submitting the above manuscript to PLOS ONE. During our internal evaluation of the manuscript, we found significant text overlap between your submission and the following previously published works, some of which you are an author.

- http://pubs.sciepub.com/ajcp/4/1/1/

Therefore, we cannot consider your manuscript as it stands. Please revise the manuscript to rephrase the duplicated text and fully cite all your sources, where appropriate.

Reviewers' comments:

Reviewer's Responses to Questions

**Comments to the Author**

1. Is the manuscript technically sound, and do the data support the conclusions?

Reviewer #1: Yes

Reviewer #2: Yes

2. Has the statistical analysis been performed appropriately and rigorously? 

Reviewer #1: Yes

Reviewer #2: Yes

3. Have the authors made all data underlying the findings in their manuscript fully available?

Reviewer #1: Yes

Reviewer #2: Yes

4. Is the manuscript presented in an intelligible fashion and written in standard English?

Reviewer #1: Yes

Reviewer #2: Yes

5. Review Comments to the Author

Reviewer #1: I have carefully read the manuscript and my opinion is that the manuscript has a merit to be published in your reputable journal with some minor corrections. The manuscript is original, informative and readable. The authors aimed to understand the dose-response relation between physical activity and cancer, as well as the associations between meeting US physical activity guidelines and overall cancer risk in US college students. This is the first epidemiological study to examine the association between physical activity and cancer using data from the American College Health Association-National College Health Assessment (ACHA-NCHA) and the feedback might have the great practical implications. The abstract is missing the description of the method precisely. Some of the experienced researchers can reach the relevant information from the method section, but I suggest authors to briefly describe the method of the study as I expect that many young researchers and students will read it, so has to be clearly understandable. The introduction is well written, while materials and methods section is very well prepared and organized according to contemporary methodological rules. At the end, I have no amendments on results and discussion part but I would recommend to the authors to prepare the separate conclusion part in the following order: the main conclusions (with practical implication), the limitations of the study (more precisely) as well as recommendations for the further studies (it is very important to briefly elaborate it and highlight the most important notes). Lastly, I would recommend you to accept this manuscript right after I confirm the authors revise it in the adequate manner.

Reviewer #2: The manuscript is very well written and the topic is quite interesting for a general audience. The methodology and content are also well done. I recommend that it be accepted with minor revisions. However, I do have some comments.

First, I suggest that in the abstract, you include only the necessary numerical values and limit their use as much as possible.

Secondly, in the introduction, in addition to listing previous studies (3-6) that have examined the given topic, it is important to highlight some relevant findings that are important for the research.

Finally, in the discussion section, I recommend that you relate the results of your study to similar studies and explain the mechanisms behind the obtained results by referring to the existing literature. Expand the discussion and provide a more detailed explanation of the mechanisms underlying the obtained results.

6. PLOS authors have the option to publish the peer review history of their article (what does this mean?). If published, this will include your full peer review and any attached files.

Reviewer #1: **Yes: **Stevo Popovic

Reviewer #2: No

---

## [Author Response · Author response to Decision Letter 0]

9 May 2023

Authors’ Responses to Editor and Reviewers’ Comments

RE: Physical Activity is Inversely Associated with Overall Cancer Risk Among College Students in the United States: Results from the National College Health Assessment

Responses to Editor

1. Editor: Please ensure that your manuscript meets PLOS ONE's style requirements, including those for file naming. The PLOS ONE style templates can be found at https://journals.plos.org/plosone/s/file?id=wjVg/PLOSOne_formatting_sample_main_body.pdf and 

Response: Thank you for your guidance. Please see the revised manuscript which meets PLOS ONE's style requirements.

2. Editor: Thank you for submitting the above manuscript to PLOS ONE. During our internal evaluation of the manuscript, we found significant text overlap between your submission and the following previously published works, some of which you are an author.

- http://pubs.sciepub.com/ajcp/4/1/1/

Therefore, we cannot consider your manuscript as it stands. Please revise the manuscript to rephrase the duplicated text and fully cite all your sources, where appropriate.

Response: Thank you for your comments. The manuscript was revised as suggested, and some duplicated text was rephrased. The reference list was reviewed, and it is complete and correct. There are no cited papers that have been retracted.

Responses to Reviewer 1

Reviewer: I have carefully read the manuscript and my opinion is that the manuscript has a merit to be published in your reputable journal with some minor corrections. The manuscript is original, informative and readable. The authors aimed to understand the dose-response relation between physical activity and cancer, as well as the associations between meeting US physical activity guidelines and overall cancer risk in US college students. This is the first epidemiological study to examine the association between physical activity and cancer using data from the American College Health Association-National College Health Assessment (ACHA-NCHA) and the feedback might have the great practical implications. The abstract is missing the description of the method precisely. Some of the experienced researchers can reach the relevant information from the method section, but I suggest authors to briefly describe the method of the study as I expect that many young researchers and students will read it, so has to be clearly understandable. The introduction is well written, while materials and methods section is very well prepared and organized according to contemporary methodological rules. At the end, I have no amendments on results and discussion part but I would recommend to the authors to prepare the separate conclusion part in the following order: the main conclusions (with practical implication), the limitations of the study (more precisely) as well as recommendations for the further studies (it is very important to briefly elaborate it and highlight the most important notes). Lastly, I would recommend you to accept this manuscript right after I confirm the authors revise it in the adequate manner.

Response: Thank you for your encouraging words and valuable comments. Your suggestions (the method of the study in the abstract and separating conclusion part) have been taken in the revised manuscript (abstract and the first paragraph of the introduction section).

Responses to Reviewer 2

Reviewer: The manuscript is very well written and the topic is quite interesting for a general audience. The methodology and content are also well done. I recommend that it be accepted with minor revisions. However, I do have some comments.

First, I suggest that in the abstract, you include only the necessary numerical values and limit their use as much as possible.

Secondly, in the introduction, in addition to listing previous studies (3-6) that have examined the given topic, it is important to highlight some relevant findings that are important for the research.

Finally, in the discussion section, I recommend that you relate the results of your study to similar studies and explain the mechanisms behind the obtained results by referring to the existing literature. Expand the discussion and provide a more detailed explanation of the mechanisms underlying the obtained results.

Response: Thank you for your encouraging words and valuable comments. The numerical values in the abstract were limited as much as possible as suggested. In the introduction, some relevant findings were highlighted. In the discussion section, although no study examined the association between PA and cancer in U.S. college students, we cited several relevant studies in the second graph and moved the mechanism section from introduction to the discussion section and provided a more detailed explanation of the mechanisms underlying the obtained results as suggested (second paragraph of the discussion section).

---

## [Decision Letter · Decision Letter 1]

31 May 2023

Physical Activity is Inversely Associated with Overall Cancer Risk Among College Students in the United States: Results from the National College Health Assessment

PONE-D-23-06157R1

Dear Dr. Wu,

We’re pleased to inform you that your manuscript has been judged scientifically suitable for publication and will be formally accepted for publication once it meets all outstanding technical requirements.

Kind regards,

Bojan Masanovic, Ph.D.

Academic Editor

PLOS ONE

Additional Editor Comments (optional):

Reviewers' comments:

Reviewer's Responses to Questions

**Comments to the Author**

1. If the authors have adequately addressed your comments raised in a previous round of review and you feel that this manuscript is now acceptable for publication, you may indicate that here to bypass the “Comments to the Author” section, enter your conflict of interest statement in the “Confidential to Editor” section, and submit your "Accept" recommendation.

Reviewer #1: All comments have been addressed

Reviewer #2: (No Response)

2. Is the manuscript technically sound, and do the data support the conclusions?

Reviewer #1: Yes

Reviewer #2: Yes

3. Has the statistical analysis been performed appropriately and rigorously? 

Reviewer #1: Yes

Reviewer #2: Yes

4. Have the authors made all data underlying the findings in their manuscript fully available?

Reviewer #1: Yes

Reviewer #2: Yes

5. Is the manuscript presented in an intelligible fashion and written in standard English?

Reviewer #1: Yes

Reviewer #2: Yes

6. Review Comments to the Author

Reviewer #1: No further requirements from my side. The authors satisfied all the requirements I requested in the initial review.

Reviewer #2: I think that the authors have put in effort and made the necessary revisions. Therefore, I propose that the work should be accepted for publication.

7. PLOS authors have the option to publish the peer review history of their article (what does this mean?). If published, this will include your full peer review and any attached files.

Reviewer #1: No

Reviewer #2: **Yes: **Borko Katanic

---

## [Editor Report · Acceptance letter]

1 Jun 2023

PONE-D-23-06157R1 

Physical Activity is Inversely Associated with Overall Cancer Risk Among College Students in the United States: Results from the National College Health Assessment 

Dear Dr. Wu:

I'm pleased to inform you that your manuscript has been deemed suitable for publication in PLOS ONE. Congratulations! Your manuscript is now with our production department. 

Kind regards, 

on behalf of

Dr. Bojan Masanovic 

Academic Editor

PLOS ONE